# Antineoplastic Properties by Proapoptotic Mechanisms Induction of *Inula viscosa* and Its Sesquiterpene Lactones Tomentosin and Inuviscolide

**DOI:** 10.3390/biomedicines10112739

**Published:** 2022-10-28

**Authors:** Rossana Migheli, Patrizia Virdis, Grazia Galleri, Caterina Arru, Giada Lostia, Donatella Coradduzza, Maria Rosaria Muroni, Giorgio Pintore, Luigi Podda, Claudio Fozza, Maria Rosaria De Miglio

**Affiliations:** 1Department of Medicine, Surgery and Pharmacy, University of Sassari, 07100 Sassari, Italy; 2Department of Biomedical Sciences, University of Sassari, 07100 Sassari, Italy

**Keywords:** *Inula viscosa*, tomentosin, inuviscolide, intrinsic pathway of apoptosis, extrinsic pathway of apoptosis, ROS production, cytotoxic effect

## Abstract

Cancer is a complex disease including approximately 200 different entities that can potentially affect all body tissues. Among the conventional treatments, radiotherapy and chemotherapy are most often applied to different types of cancers. Despite substantial advances in the development of innovative antineoplastic drugs, cancer remains one of the most significant causes of death, worldwide. The principal pitfall of successful cancer treatment is the intrinsic or acquired resistance to therapeutic agents. The development of more effective or synergistic therapeutic approaches to improve patient outcomes and minimize toxicity has become an urgent issue. *Inula viscosa* is widely distributed throughout Europe, Africa, and Asia. Used as a medicinal plant in different countries, *I. viscosa* has been characterized for its complex chemical composition in order to identify the bioactive compounds responsible for its biological activities, including anticancer effects. Sesquiterpene lactones (SLs) are natural, biologically active products that have attracted considerable attention due to their biological activities. SLs are alkylating agents that form covalent adducts with free cysteine residues within enzymes and key proteins favoring cancer cell cytotoxicity. They are effective inducers of apoptosis in several cancer cell types through different molecular mechanisms. This review focuses on recent advances in the cytotoxic effects of *I. viscosa* and SLs in the treatment of neoplastic diseases, with a special emphasis on their proapoptotic molecular mechanisms.

## 1. Introduction

Traditional herbal medicines are excellent sources of natural biologically active products with powerful therapeutic effects that can be utilized in the treatment of various diseases, including cancer. *Inula viscosa* (L.) Aiton. (*Dittrichia viscosa* L.) is a medicinal perennial plant belonging to the genus *Inula* and family *Asteraceae* and is considered one of the most crucial pharmaceutical plants in the Mediterranean Basin [1]. As a medicinal plant in, *I. viscosa* has been characterized in different countries for its complex chemical composition to identify the bioactive compounds responsible for its biological activities [2]. *I. viscosa* has long been used in folk medicine owing to its anti-inflammatory [3], anthelmintic, antipyretic, antiseptic, and antiphlogistic activities [4,5], and in the treatment of lung disorders [6] and diabetes [7]. Hernandez et al. isolated and tested several flavanones, such as sakuranetin, 7-O-methylaromadendrin, and 3-acetyl-7-O-methylaromadendrin showing relevant effects on enzymes involved in the inflammatory response [8]. Recently, several studies have explained the biological mechanisms triggering the anti-inflammatory effect of *I. viscosa*, which are based on the inhibition of COX1, COX2, and iNOS enzymatic activity [9,10]. Kheyar-Kraouche et al. performed a high-performance liquid chromatography associated with electrospray ionization mass spectrometry on the ethanolic extract obtained by *I. viscosa* leaves growing in Algeria. They identified different phytochemical components, including phenolic acids, flavonoids, lignans, and terpenoids; some of which were recognized for the first time and belong to different subfamilies of compounds [11].

Recently, cytotoxic and anticancer activities of *I. viscosa* have been demonstrated in various cancer cell lines [12,13,14]. Therefore, elucidating the anticancer properties of this plant could be a relevant strategy to identify new antitumor agents. Different biologically active compounds belonging to several families isolated from *I. viscosa* were tested alone or in association with cancer cell lines [15,16,17,18], and in vivo experiments showed their strong antineoplastic activity [19,20]. Messaoudi et al. performed a chemical composition analysis of different *I. viscosa* extracts from three different regions of Morocco, revealing the presence of sesquiterpene lactones (SLs), tomentosin, and inuviscolide as major representative compounds, with tomentosin concentrations ranging from 22% to 64% and inuviscolide concentrations ranging from 0% to 58% in different Moroccan regions [14]. The SL structure consists of 15-carbon terpenoids obtained by the condensation of three isoprene units and a lactone ring [21]. Most sesquiterpenes, but not all, contain the α-methylene-γ-butyrolactone motif, the functional group responsible for their biological effects, and, above all, the antitumor activity [22]. Although they are mainly found in the *Asteraceae* family, SLs have been identified in several families of flowering plants, including *Solanaceae*, *Araceae*, and *Cactaceae* [23]. As reported by Wang et al., 396 types of sesquiterpenoids with high structural diversity have been isolated and characterized with the *Inula* (*Asteraceae*) genus [2].

Recently, several studies have reported the potential anticancer effects of SLs and have shown that tomentosin and inuviscolide exert antiproliferative effects on human cancer cell lines [12,24]. In particular, SLs are alkylating agents that cause DNA damage with consequent activation of ATM/ATR kinase and involvement of CDC2, TP53, survivin, and NF-κB inducing cell cycle arrest and apoptosis activation [12]. Tomentosin and inuviscolide, purified from *I. viscosa* extracts, characterize previous molecular mechanisms. SLs induce cell cycle arrest in the G2/M phase and appearence of a sub-G0 fraction evocative of apoptotic cell death in melanoma cell lines [12]. Japonicone A, derived from *I. japonica*, performs antineoplastic activity against Burkitt lymphoma (BL) cells [25]. Additionally, studies by Merghoub et al. identified tomentosin as responsible of antineoplastic effects of *I. viscosa* on cervical cancer cells [26].

This review summarizes the emerging roles of *I. viscosa* and its prevalent SLs in the treatment of neoplastic diseases, with special emphasis on their proapoptotic molecular mechanisms.

## 2. Materials and Methods

The purpose of this review is to provide an overview of the current knowledge regarding the antineoplastic role of *I. viscosa* and SLs in solid and hematologic cancers. A literature search for original and review articles was performed using electronic databases of Medline (PubMed, PubMed Central) by using the terms ‘*Inula viscosa*’, ‘tomentosin’, ‘inuviscolide’, ‘cancer’, ‘carcinoma’, ‘apoptosis’. Combinations of these terms were used to screen the mentioned databases for relevant content: title, abstract, and full content, respectively. Only studies written in English were considered for evaluation. Pre-screening and screening selection removed duplicate studies, foreign language studies, irrelevant studies, and studies for which updated research was unavailable. We considered both clinical and experimental studies (in vivo and in vitro).

## 3. Results

### 3.1. Inula viscosa: Antineoplastic Activities and Molecular Mechanisms

Cytotoxic screening models represent crucial preliminary data for analyzing the antineoplastic properties of selected plant extracts. Early tests were cell-based assays performed on established cell lines in which the toxic effects of plant extracts or isolated compounds could be measured. Conventional antitumor agents display significant cytotoxic activities in cell culture systems [27].

Benbacer et al., with the purpose of developing new anticancer drugs against cervical cancer, applied the human cervical carcinoma SiHa and HeLa cell lines as a model system to screen the anticancer effects of plants from traditional Moroccan medicine. In particular, they demonstrated that *I. viscosa* hexane extract showed pronounced cytotoxic effects against both cervical cancer cell lines, inducing dose-dependent cell growth inhibition by stimulating apoptosis, which is related to the decrease in mitochondrial membrane potential (ΔΨ*m*) and increase in intracellular reactive oxygen species (ROS) production. Thus, *I. viscosa* extracts showed significant cytotoxic effects against cervical cancer cell lines through the inhibition of proliferation and induction of apoptosis involving a mitochondria-mediated signaling pathway by pro-caspase activation, *BCL-2* expression, and PARP cleavage [13]. The same group demonstrated that *I. viscosa* extracts target the telomerase machinery and induce apoptosis in human cervical carcinoma cell lines (SiHa and HeLa) and that the molecular mechanism underlying *I. viscosa* extract-induced apoptosis includes a caspase-3 mediated signaling pathway [28].

An interesting study by Messaoudi et al. showed the cytotoxic activity of ethyl acetate and ethanolic *I. viscosa* extracts harvested from three different regions of Morocco (Taouante, Sefrou, and Imouzzer) in two breast cancer cell lines, MCF7, an estrogen receptor-positive cell line, and MDAMB-231, an estrogen receptor-negative cell line. These two *I. viscosa* extracts showed different toxicity on breast cancer cells, suggesting that the different cytotoxic activity can be an integral effect of the combination of three major compounds, tomentosin, inuviscolide, and isocostic acid which are present in variable concentrations in plants from different regions. Furthermore, the reduced toxicity exerted by the two extracts on MCF-7 cells when compared with MDA-MB-231 cells suggests that heterogeneous susceptibility could be dependent on the activation of different signaling pathways [14]. In addition, they demonstrated that ethanolic and ethyl acetate extracts of *I. viscosa* from Taounate, Imouzzer, and Sefrou had different rates of polyphenols associated with different antioxidant activities [29]. Further studies have demonstrated the selective cytotoxic effects of *I. viscosa* on MCF-7 cells [30,31].

Bar-Shalom et al. examined the possible therapeutic effects of *I. viscosa* aqueous extract on colon cancer cell growth in vitro and tumor growth in vivo, using a xenograft mouse model. In vitro experiments revealed that exposure of colorectal cancer cells to *I. viscosa* extract significantly reduced cell viability in a dose- and time-dependent manner. Interestingly, the analysis of the molecular mechanisms underlying the *I. viscosa* effect showed the activation of caspase-9 in HCT116 well-differentiated cells and of caspases-8 and -9 in colo320 poorly-differentiated cells. These findings suggest that *I. viscosa* extract induces apoptosis through the intrinsic mitochondrial pathway in well-differentiated cells, and through both the intrinsic and extrinsic pathways in poorly-differentiated cells. In vivo studies revealed that treatment with *I. viscosa* extract inhibited tumor growth in mice transplanted with MC38 cells, showing a strong reduction in the weight and volume of neoplastic lesions. Interestingly, no side effects such as weight loss, behavioral changes, ruffled fur, or changes in kidney and liver function were observed, suggesting the absence of toxicity from *I. viscosa* [32].

*I. viscosa* collected from an uncontaminated area of the National Park on Asinara Island, Sardinia, revealed powerful anti-lymphoma activity. Specifically, Raji cells treated with increasing concentrations of *I. viscosa* ethanolic extract demonstrated a dose- and time-dependent decrease in cell viability, displaying a reduction in cell proliferation obtained by the induction of cell cycle arrest in the G2/M phase, and a dose-dependent increase in cell apoptosis. A gene expression analysis of signal transduction and apoptotic pathway players involved in B-lymphocyte functions showed that the molecular mechanisms involved in *I. viscosa* anticancer activity were characterized by the downregulation of genes involved in cell cycle and proliferation (*c-MYC*, *CCND1*), as well as in the inhibition of cell apoptosis (*BCL2*, *BCL2L1*, *BCL11A*) [33].

An interesting study evaluated the cytotoxic and anticancer effects of *I. viscosa* methanol and aqueous extracts on the malignant melanoma cell lines A2058 and MeWo, and normal fibroblasts. Cytotoxicity, apoptosis induction, and migration suppression were strongly induced in malignant melanoma cell lines by *I. viscosa* methanol extracts compared to the aqueous extracts [34], confirming that the solvent used in the extraction steps can influence the content and biological activity of the extract [28,30]. For the first time, an epigenetic mechanism underlying the anticancer activity of *I. viscosa* has been demonstrated. Specifically, *I. viscosa* methanol extract promotes the downregulation of miRNAs related to epithelial-mesenchymal transition and poor prognosis in malignant melanoma, such as miR-191 and miR-193, while favoring the overexpression of miR-579 and miR-524, which mainly repress the MAPK signaling pathway in malignant melanoma [34].

The ubiquitin–proteasome system plays a key role in intracellular proteolysis, particularly in the degradation of abnormal proteins. In fact, it is directly involved in the regulation of most biological processes, such as cell cycle, apoptosis, muscle differentiation, and immune response [35]. Many studies have revealed that proteasome levels can be used as biomarkers for various types of cancer [36,37,38]. Recently, Yaagoubi et al. investigated the antitumor effects and proteasome inhibition capacity of *I. viscosa* extract in a mouse model of DMBA/croton oil-induced skin carcinoma. Animals received treatment with the extract before and after the induction of skin carcinogenesis, showing that *I. viscosa* extract inhibited the development of papilloma in mice. Furthermore, ingestion of *I. viscosa* extract delayed the formation of cutaneous papillomas in animals and simultaneously decreased the size and number of papillomas. A structure–activity study showed that *I. viscosa* extract contains bioactive molecules with much greater inhibition of the subunits of the proteasome, as well as a decrease in the concentration of proteasome and its catalytic activity in serum and intracellularly when compared to chemically synthesized inhibitors, thus emerging as a new candidate for targeted therapy against skin carcinoma. Specifically, molecular docking analysis revealed that tomentosin, inuviscolide and isocostic acid compounds obtained from *I. viscosa* extract were stabilized in the pocket of the 20S proteasome β5 receptor subunits by various interactions, mirroring the same mechanisms exerted by carfilzomib, a potent second-generation proteasome inhibitor with significant anti-myeloma activity [39]. *I. viscosa* extracts also increased cell cycle arrest and cell death in the glioblastoma LN229 cell line, characterized by a TP53 mutation, compared to U87MG cells with wild-type TP53. SW620 cells were more sensitive to *I. viscosa* extracts, suggesting that they may contain molecules with high therapeutic potential against MDR cell lines. PC-3 are androgen-insensitive and apoptosis-resistant prostate cancer cells, on which *I. viscosa* extracts induce growth inhibition, cell cycle arrest and apoptosis, supporting the idea that active compounds effectively targeting extrinsic and intrinsic apoptosis pathways are present in the plant [40]. Table 1 summarizes the biological and molecular effects induced by *I. viscosa* treatment with in vitro and in vivo models.

### 3.2. Sesquiterpene Lactones Tomentosin and Inuviscolide: Antineoplastic Activities and Molecular Mechanisms

Different biologically active compounds, belonging to several families, isolated from *I. viscosa* were tested alone or in association in cancer cell lines [15,16,17,18] or within in vivo experiments [19,20], showing strong antineoplastic activity. SLs represent one of the most abundant and globally distributed groups of plant-derived bioactive compounds and have been identified as worthwhile therapeutic agents against various types of cancer [21,22].

In SLs, it is assumed that the electrophilic ab-unsaturated carbonyl structures, such as α-methylene-γ-lactone, represent bioactive functional groups interacting in a Michael-type addition with the nucleophilic sites of biological molecules. SLs are thought to inhibit tumor cell proliferation by selective alkylation of cysteine sulfhydryl groups in growth-regulatory biological macromolecules, such as key enzymes that control cell division by restoring the ability of tumor cells to undergo apoptosis [41]. In addition, DNA alkylation is a potential molecular cytotoxicity mechanism of SLs [42]. SLs are potentially selective toward neoplastic and cancer stem cells by targeting specific signaling pathways, making them lead compounds in cancer clinical trials [43]. Although the specific molecular targets and mechanisms of the antitumor activity of SLs in vitro and in vivo have not yet been explained, different studies have reported that SLs, tomentosin and inuviscolide, exert antiproliferative and proapoptotic effects on various human cancer cell lines, then trying to identify the underlying molecular mechanisms.

Rozenblat et al. purified the SLs, tomentosin and inuviscolide, from *I. viscosa* leaves and analyzed their anticancer effectiveness against human melanoma cell lines with the aim of developing new agents for melanoma treatment, taking into consideration the aggressiveness and chemotherapeutic resistance of this tumor. Tomentosin and inuviscolide reduced cell viability in three different human melanoma cell lines by favoring cell cycle arrest at the G2/M phase, associated with an increase in the sub-G0 fraction indicative of apoptotic cell death, as demonstrated by changes in membrane phospholipids, mitochondrial membrane potential, and activation of caspase-3. The molecular mechanism of SL-mediated G2/M arrest and apoptosis in melanoma cell lines suggests that SLs are possible alkylating agents that might cause DNA damage, which activates the kinase ATM/ATR. This activation is followed by early phosphorylation of TP53 (Ser15) and CDC2 (Thr14 and Tyr15) providing early G2/M arrest. The activation of TP53 transactivates p21waf1, which also reduces the protein concentration of CDC2/Cyclin B1, favoring the elongation of G2/M arrest, and ultimately resulting in apoptosis [12]. To better understand the mechanisms responsible for apoptotic death, the authors detected the effects of SLs on survival protein, such as the survivin that favor the acquisition of chemoresistance of neoplastic cells by its anti-apoptotic potency based on the inhibition of the effector caspase 3 and 7 [44]. In human melanoma cell lines, the decreased levels of survivin (by both SLs) and the p65/RelA subunit of NF-κB (only by inuviscolide) suggest an apoptotic induction. The induction of apoptosis by tomentosin and inuviscolide in aggressive human melanoma cell lines has high pharmacological value, implying that SLs could be potentially developed as novel agents for melanoma treatment [12].

To explain possible mechanisms underlying tomentosin-induced apoptosis, Merghoub et al. studied the effects of tomentosin on telomere lengthening by hybridization with a telomeric C-rich probe (21C) under non-denaturing conditions, which drastically induced telomere G-overhang shortening in human cervical cell lines. This mechanism was validated on JW10 cells in which tomentosin induced a significant anti-proliferative effect acting on hTERT expression, while it exhibited a low cytotoxic effect on Wi38 fibroblast cells, a primary cell culture without telomerase expression and activity [26]. Furthermore, the molecular mechanism underlying tomentosin-induced apoptosis in human cervical cell lines involves a mitochondria-mediated signaling pathway. In fact, tomentosin obtained from the aerial parts of *I. viscosa* causes a reduction in the mitochondrial membrane potential and an increase in ROS levels in human cervical cell lines. It leads to the downregulation of pro-caspase-3 protein, cleavage of PARP, and enhanced caspase-3 activity associated with *BCL2* downregulation in tomentosin-treated SiHa and HeLa cells [26].

ROS are commonly produced by mitochondrial oxidative metabolism in response to cellular stress [45]. They are vital chemical messengers that play essential roles in cell growth and proliferation [46]. Paradoxically, the pro-oxidant activity of phytochemicals has been described as a critical mechanism underlying their anticancer effects [47]. In fact, celastrol causes G2/M cell cycle arrest, autophagy, and apoptosis through the ROS/JNK pathway in human osteosarcoma cells [48], and phenyl arsine oxide induces apoptosis in HepG2 cells via ROS-dependent signaling pathways [49]. Lee et al. analyzed the role of intracellular ROS in tomentosin-induced apoptosis in an osteosarcoma cell line and demonstrated that the tomentosin-induced apoptosis can be inhibited through the suppression of ROS production by N-acetyl-cysteine. Moreover, a decrease in peroxiredoxin-1, an antioxidant enzyme that reduces the levels of hydrogen peroxide and alkyl hydroperoxides, has been shown after treatment with tomentosin. These results support the hypothesis that tomentosin-induced apoptosis is associated with intracellular ROS production. By analyzing the molecular mechanisms, the authors demonstrated that tomentosin-induced ROS upregulate *FOXO3* and *p27* expression, thus suggesting that *FOXO3* upregulation controls G2/M phase cell cycle arrest through *p27* overexpression after tomentosin treatment in an osteosarcoma cell line [24]. A similar molecular mechanism was identified by Yu et al., who investigated the role of tomentosin in hepatocellular carcinoma cell lines (HepG2 and Huh7), in which tomentosin induced G2/M phase cell cycle arrest through *p27* overexpression regulated by the upregulation of *FOXO3*. In addition, cell cycle arrest and apoptosis induction are boosted by the overexpression and phosphorylation of TP53 and activation of the ERK signaling pathway [50].

A recent study validated the hypothesis that tomentosin-induced oxidative stress is involved in apoptosis via the mitochondria-mediated signaling pathway in gastric carcinoma cell lines. The molecular mechanisms that explain the antiproliferative effect of tomentosin are based on the reduction in PCNA and cyclin D1, which are able to control cell growth. The induction of apoptosis is related to *BAX* overexpression, *BCL-2* downregulation, and inhibition of inflammation, as revealed by the downregulation of IL-6, TNF-α, IL-1β, and IL-8 which modulate cell growth proteins in gastric cancer cells [51].

Recently, the antineoplastic activity of SLs has also been evaluated in hematologic tumors such as LB, multiple myeloma (MM), and leukemia cancer cell lines. Virdis et al. demonstrated that tomentosin exerts strong antitumor activity on human BL (Raji cell line; [52]) and MM (RPMI 8226 cell line; [53]), mediated by inhibition of cell proliferation and induction of apoptosis. Apoptosis was induced by activating both the death receptor and mitochondrial pathways in Raji cells. Gene expression profiling analysis was performed to assess differentially expressed genes contributing to tomentosin activity in BL and MM. Seventy-five genes deregulated by tomentosin were identified in BL. Downregulated genes are enriched in the immune system, PI3K/AKT, and JAK/STAT pathways, which assist proliferation and growth. Notably, different deregulated genes identified in tomentosin-treated BL cells are prevalent in molecular pathways known to lead to cellular death, downregulation of anti-apoptotic genes such as *BCL2A1* and *CDKN1A*, and upregulation of the proapoptotic *PMAIP1* gene [52]. In total, 126 genes deregulated by tomentosin were identified in MM. In total, 126 genes deregulated by tomentosin were identified in MM. Protein–protein interaction network analysis revealed that the tomentosin treatment of MM produced the downregulation of genes involved in pathways implied in immune system processes and in cellular neoplastic processes, such as growth, proliferation, migration, invasion, and apoptosis. Furthermore, tomentosin causes endoplasmic reticulum stress via upregulation of the *ATF4* and *DDIT3* genes, suggesting that tomentosin treatment the activation of the protective unfolded protein response signaling might induce cell apoptosis. Functional connection analysis executed by the connectivity map tool indicated that tomentosin acts as the heat shock protein inhibitors on MM cells [53]. 

Furthermore, Yang et al. showed that tomentosin stimulates intracellular ROS production, causing mitochondria-centered death in leukemia MOLT-4 cells, revealing significant cytotoxicity. Tomentosin induces apoptosis in MOLT-4 cells by suppression of NF-κB and proinflammatory cytokines, as revealed by the complete blocking of anti-apoptotic proteins (cyclin D1 and BcL-2) and activation of proapoptotic proteins (caspase-3 and BAX) via the mTOR/PI3K/AKT pathway [54]. These findings suggest that tomentosin could be considered a potential natural product with limited toxicity and relevant antitumor activity among the therapeutic options available for onco-hematologic patients.

Recently, Güçlü et al. analyzed the anticancer effects of tomentosin on PANC-1 and MIA PaCa-2 human pancreatic cancer cells on which the treatment induces suppression of proliferation, migration, invasion, and colony formation capacity. Tomentosin increases apoptosis rate and ROS production and decreases mitochondrial membrane potential apoptosis in pancreatic cancer cells. At a molecular level, tomentosin induces overexpression of apoptosis-related genes, such as *CASP8*, *PPARG*, *FAS*, *FADD*, *TNF*, and *TNFR1* genes for extrinsic pathway and *BAX*, *BCL2*, *CASP3*, *CASP7*, *CASP9*, and *CYCS* genes for intrinsic pathway and increases caspase-3 and caspase-9 protein levels [55]. 

Table 2 summarizes the biological and molecular effects induced by SLs treatment within in vitro system models. Figure 1 describes the prevalent proapoptotic molecular mechanisms induced by SLs treatment in human tumors.

## 4. Conclusions and Future Prospects

Despite the advances in the development of innovative antineoplastic drugs, cancer remains one of the most important causes of death worldwide. The principal pitfall of successful cancer treatment is the intrinsic or acquired resistance to therapeutic agents. Among the conventional treatment options, chemotherapy is most often applied to different types of cancer. Several chemotherapeutic molecules induce cell cycle arrest but not apoptosis, permitting neoplastic cells to repair their damaged DNA and therefore potentially limit treatment effectiveness [26]. These approaches can be associated with variable response rates and potentially severe side effects, which may impair quality of life and often favor cancer progression. Thus, the development of more effective and/or synergistic anticancer agents that can overcome resistance and are not associated with severe side effects has become an extremely important issue. Recently, drugs belonging to the glycolytic inhibitor category have attracted considerable attention due to their capacity of inhibiting aerobic glycolysis which represents an attractive strategy to specifically kill tumor cells. Interestingly, small molecule alkylating agents have shown great potential, such as 3-bromopyruvate as a promising antitumor drug by blocking tumor energy metabolism [56], as well as lonidamine that may be very promising as a sensitizer of tumors to chemotherapeutic agents and physical therapies [57].

This review describes recent advances in our understanding of the potential anticancer activity of *I. viscosa* and its natural biologically active compounds, such as tomentosin and inuviscolide. Moreover, it provides a detailed description of the molecular mechanisms of action in the context of in vitro experiments. Interestingly, cytotoxic effects against human cancer cell lines induced by *I. viscosa* and SLs are characterized by cell growth inhibition and apoptosis induction, related to the decrease in mitochondrial membrane potential and increase in intracellular ROS production. Various molecular mechanisms are responsible for the activation of apoptosis. In particular, cell treatment induces activation of pathways known to lead to cellular death, downregulation of anti-apoptotic genes such as BCL2A1 and CDKN1A, and upregulation of the proapoptotic PMAIP1 gene. In addition, SLs can target multiple signaling pathways, such as NF-κB, PIK/AKT/mTOR, and MAPK, which control apoptosis, growth, proliferation, migration, and invasion. In vivo tests within preclinical animal studies and subsequently in the context of controlled clinical trials should also be encouraged to shed further light on the exact molecular mechanisms by which tomentosin induces pharmacological effects in human cancer and to assess possible side effects. These preliminary steps could allow the development of potential natural products with limited toxicity and relevant antitumor activity, making them available among the therapeutic armamentarium offered to patients affected by solid and hematologic tumors.

## Figures and Tables

**Figure 1 biomedicines-10-02739-f001:**
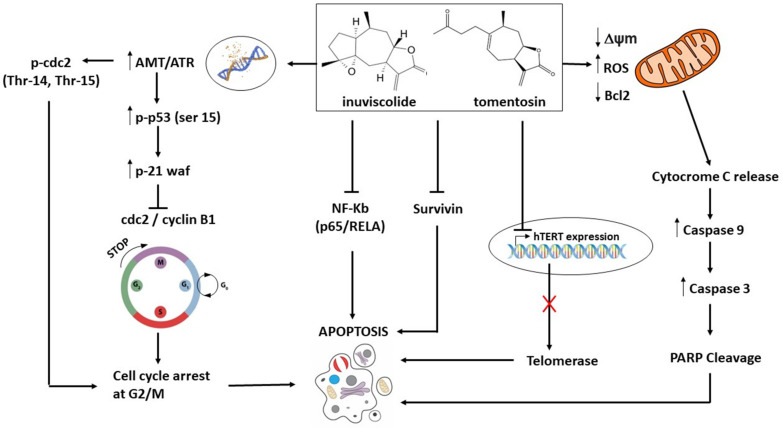
Molecular mechanisms of apoptosis induced by tomentosin and inuviscolide in neoplastic cells.

**Table 1 biomedicines-10-02739-t001:** Summary of biological and molecular effects induced by *Inula viscosa* treatment with in vitro and in vivo system models.

Type of Cancer	Treatment	Model	Biological and Molecular Effects	Ref.
Cervical cancer	Hexane extract from 15.6 to 500 μg/mL for 48 or 72 h	SiHa and HeLa cell lines	Cell growth inhibition and apoptosis induction. Decrease ΔΨm and intracellular ROS production	[13]
Cervical cancer	Hexane and methanol extract from 5 to 80 μg/mL for 72 h	SiHa and HeLa cell lines	Cell growth inhibition and apoptosis induction. Inhibition of telomerase activity and induction of telomere shortening	[28]
Breast cancer	Ethyl acetate and ethanolic extract from 15.6 to 500 μg/mL for 72 h	MCF-7 and MDA-MB231 cell lines	Cytotoxic effect	[14]
Colorectal cancer	Aqueous extract from 100 to 300 μg/mL for 24–72 h	HCT116 and colo320 cell lines	Reduction in cell viability. Apoptosis induction through the intrinsic mitochondrial pathway in well-differentiated cells and through both, the intrinsic and extrinsic pathways in poorly differentiated cells	[32]
	Aqueous extract 150 or 300 mg/kg	C57BL/6 mice transplanted with MC38 cells	Inhibition of tumor growth	
Burkitt lymphoma	Ethanolic extract: 5, 10, 20, 30, 40, 60, and 80 mg/mL for 24 and 48 h	Raji cell line	Cell cycle arrest in the G2/M phase, decreased cell viability and increased cell apoptosis. Downregulation of genes involved in cell cycle and proliferation (*c-MYC*, *CCND1*) and in the inhibition of cell apoptosis (*BCL2*, *BCL2L1*, *BCL11A*)	[33]
Malignant melanoma	Aqueous and methanolic extracts from 10 µg/mL to 140 µg/mL for 24–72 h	A2058 and MeWo cell lines	Antiproliferative effect by induction of apoptosis and cell cycle arrest, suppression of cell migration. Deregulation of oncogenic and oncosuppressive miRNAs	[34]
Skin carcinogenesis	Ethanolic extract 100 μL for 4 days	Swiss albino mice treated with DMBA/croton oil	Inhibition of the development of papilloma. *I. viscosa* extract induces inhibition on the subunits of the proteasome, as well as decrease in the concentration of proteasome and its catalytic activity in serum and intracellularly.	[39]

**Table 2 biomedicines-10-02739-t002:** Summary of the biological and molecular effects of sesquiterpene lactones treatment with in vitro system models.

Type of Cancer	Treatment	Model	Biological and Molecular Effects	Ref.
Melanoma	Tomentosin and Inuviscolide 9–36 mM for 24 h	SK-28, 624 mel, and 1363 mel cell lines	Cell cycle arrest at the G2/M phase and apoptosis. Activation of ATM/R followed by phosphorylation of TP53 and CDC2 and p21waf1 overexpression. Decrease of Survivin and of NF-kB	[12]
Cervical cancer	Tomentosin 0–100 mM for 24, 48, 72, and 96 h	HeLa and SiHa cell lines	Cell cycle arrest and apoptosis. Increased ROS and decrease in mitochondrial membrane potential. Telomeric G-overhang shortening	[26]
Osteosarcoma	Tomentosin 0, 10, 20, and 40 µM for 24 and 48 h	MG-63 cell line	Decreased cell viability and migration, apoptosis, cell cycle arrest. Increase of ROS induces *FOXO3* and *p27* overexpression. Decrease of peroxiredoxin-1	[24]
Gastric cancer	Tomentosin from 5 to 30 μM for 24 h	AGS cell line	Apoptosis via mitochondria-mediated signaling pathway induced by increase of ROS. PCNA and Cyclin D1 downregulation. *BAX* overexpression and *BCL-2* downregulation. Inhibition of inflammation	[51]
Hepatocellular carcinoma	Tomentosin 0,10, 20 and 40 μM for 24 and 48 h	HepG2 and Huh7 cell lines	G2/M phase cell cycle arrest through *p27* overexpression regulated by upregulation of *FOXO3*	[50]
Leukemia	Tomentosin 0–25 μM for up to 48 h	MOLT-4 cell line	Apoptosis via mitochondria-mediated signaling pathway induced by increase of ROS. Suppression of NF-κB and proinflammatory cytokines. mTOR/PI3K/AKT pathway activation	[54]
Burkitt lymphoma	Tomentosin 50, 25, 12.5, 6.25, 3.125, 1.56 and 0.75 μM for 24 h	Raji cell line	Cell proliferation inhibition and cell apoptosis induction. Induction of apoptosis by upregulation of the PERK/eIF2a/ATF4/DDIT3 pathway	[52]
Multiple myeloma	Tomentosin 50, 25, 12.5, 6.25, 3.125, 1.56 and 0.75 μM for 24 h	RPMI-8226 cell line	Cell proliferation inhibition and cell apoptosis induction. Downregulation of anti-apoptotic genes such as *BCL2A1* and *CDKN1A* and upregulation of the proapoptotic *PMAIP1* gene	[53]

## Data Availability

Not applicable.

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
