# Peer review of "Antineoplastic Properties by Proapoptotic Mechanisms Induction of Inula viscosa and Its Sesquiterpene Lactones Tomentosin and Inuviscolide"

_biomedicines, 2022, doi:10.3390/biomedicines10112739_

Round 1

Reviewer 1 Report

Review comments

In this review, the authors described the antitumor activities and the mechanisms of action of Inula Viscosa and its sesquiterpene lactones. Overall, this paper is worthy of publication, however, major revision is necessary. Some suggestions are listed as follows:

Some suggestions are listed as follows:

1、 In the introduction or discussion section, the authors can cite and discuss the recommended papers related to tumor energy metabolism in cancer treatment, “Cancers, 2020, 12, 3332; Cancers, 2019, 11, 317.”

2、 The first paragraph at the beginning of result 1 can be considered to be left out or put into the discussion, it is not appropriate to write in the antineoplastic activities.

3、 If I understand correctly, line 211 should be result 2, but the heading is not preceded by subheading “3.2”.

4、 Perhaps the two results sections could be divided into more details, so that the anti-tumor activities and molecular mechanisms of I. viscosa and SLs can be understood at a glance, and their mechanisms can be considered to be written separately.

5、 A large part of this review is devoted to the anti-tumor effects of I. viscosa and SLs, the molecular mechanisms of these compounds can be further explored in this paper.

6、 The results in the review can be summarized in more aspects, such as whether I. viscosa or SLs can act synergistically with some other factors to make the idea more open. 

7.  In figure 1, alkilant agents? What is it?

Author Response

We are grateful for your consideration of this manuscript, and we thank the reviewer who have permitted us to improve the quality of our work with their suggestions. All the comments we received on this study have been considered and we have addressed all of them in our point-by-point reply below.

  1. In the introduction or discussion section, the authors can cite and discuss the recommended papers related to tumor energy metabolism in cancer treatment, “Cancers, 2020, 12, 3332; Cancers, 2019, 11, 317.”

Thanks for the reviewer’s suggestions. We discussed the recommended papers related to tumor energy metabolism in cancer treatment in the conclusions.

2、 The first paragraph at the beginning of result 1 can be considered to be left out or put into the discussion, it is not appropriate to write in the antineoplastic activities.

Accordingly, with the reviewer’s comment, we have moved the first paragraph at the beginning of result 1 into the conclusions.

3、 If I understand correctly, line 211 should be result 2, but the heading is not preceded by subheading “3.2”.

Thanks for the reviewer’s comment. We added the subheading “3.2” at the heading of the result 2.

4、 Perhaps the two results sections could be divided into more details, so that the anti-tumor activities and molecular mechanisms of I. viscosa and SLs can be understood at a glance, and their mechanisms can be considered to be written separately.

Thanks for the reviewer’s comment. Considering that Inula viscosa and its sesquiterpenes have been tested in different tumor models in vitro and also in vivo, we preferred to describe their antitumor activity and molecular mechanism separately for each of these. But above all to underline how each tumor activates different molecular pathways while responding to the same drug.

5、 A large part of this review is devoted to the anti-tumor effects of I. viscosa and SLs, the molecular mechanisms of these compounds can be further explored in this paper.

Thanks for the reviewer’s comment. At the moment the effects of Inula viscosa and its sesquiterpenes have been mainly tested for cytotoxic activity towards cancer. There are not many studies that analyzed the molecular mechanisms, especially those activating apoptosis. At present, we have discussed all the published studies related to the molecular mechanisms of Inula viscosa and sesquiterpenes, which are the subject of the review.

6、 The results in the review can be summarized in more aspects, such as whether I. viscosa or SLs can act synergistically with some other factors to make the idea more open.

Thanks for the reviewer’s comment. Currently there are not studies on the cancer cell effects of the combination of Inula or sesquiterpenes with other types of antineoplastic drugs. The only analyzed combinations concern various products obtained from the Inula plant. The latter have been included in the review, i.e. Messaoudi et al. European J Med Plants 2016, 16, 1–9 (references 14).

7. In figure 1, alkilant agents? What is it?

Alkilant agents refer to tomentosin and inuviscolide, but in fact it can be confusing. So we preferred to eliminate it.

Reviewer 2 Report

The review summarizes the knowledge on Inula viscosa extracts and isolated compounds anticancer activity. This is very interesting review, however there are some aspects that requires clarification:

1)     All Latin names of plant species and genus should be written using Italic font;

2)     In the title is should be Inula viscosa instead of InulaViscosa;

3)     English grammar requires correction;

4)     Abstract: there is no need to put the short form of Inula viscosa after the full name, it is common to write full name at first, after the short form is used;

“adducts in vivo” – adducts to what?

5)     Introduction: the synonym name of Inula viscosa should be given;

“7-O-methylaromadendrin, and 3-acetyl-7-Omethylaromadendrin” – please correct;

“SLs” – the abbreviation when used for the first time should be explained; it was used in Abstract but Abstract and min text are independent parts;

“ranging from..” – what ranging?

“14”?

Guo-Wei at al ..[21] - this reference refers to Xiang, P et al. not Guo-Wei et al.

6)     Results:

Lines 130: “I. viscosa ethyl acetate and ethanol extracts display different chemical compositions and are quite different from those reported for I. viscosa grown worldwide..” - it is not surprising but where lays this difference? More information should be given;

Section between lines 131-141: this section requires clarification, it is confusing; the statements on effectiveness of ethanolic and ethyl acetate extracts on MCF-7 and MDA-MB-231 cell lines are too general;

Table 1: there is no need to repeat the name of the plant in the body of the table, this is given in the table caption;

and the first column: the references are given so this column should be removed, it does not provide any new information;

Line 214: “tested alone or in association with cancer cell lines..” - what does it mean?

Line 219: “a-b-unsaturated” and “a-methylene-g-lactone,..” – please correct;

Lines 276-279: “silencing ROS production” and “apoptosis is associated with intracellular ROS production” - this is contradictory: silencing ROS production and apoptosis caused by ROS production;

Table 2: the first column: the references are given so this column should be removed, it does not provide any new information;

Figure 1: there is no need to write names of compounds using Capital letters;

Lines 352-353: “induced by cell treatment.” – please remove;

Author Response

We are grateful for your consideration of this manuscript, and we thank the reviewer who have permitted us to improve the quality of our work with their suggestions. All the comments we received on this study have been considered and we have addressed all of them in our point-by-point reply below.

  1. All Latin names of plant species and genus should be written using Italic font;

Accordingly with the reviewer’s comment, we have put in Italic font all latin names of plant species and genus.

  1. In the title is should be Inula viscosa instead of InulaViscosa;

We modified the title as suggested.

  1. English grammar requires correction;

According with reviewer comment, we have proofread the manuscript again and experienced scholarly writers have edited this manuscript and are confident about the grammar

  1. Abstract: there is no need to put the short form of Inula viscosa after the full name, it is common to write full name at first, after the short form is used;

As suggested we eliminated the short form of Inula viscosa after the full name in the abstract.

  1. “adducts in vivo” – adducts to what?

We are grateful for the comment and agree with the reviewer for this suggestion. We rewrote the sentence to make it clearer (see lanes 26-27).

  1. Introduction: the synonym name of Inula viscosa should be given;

We added the synonym name of Inula viscosa (see lane 37) as suggested.

  1. “7-O-methylaromadendrin, and 3-acetyl-7-Omethylaromadendrin” – please correct;

We corrected the name of compounds.

  1. “SLs” – the abbreviation when used for the first time should be explained; it was used in Abstract but Abstract and min text are independent parts;

Thanks for the reviewer’s comment. We have also added the full name of the SLs in the introduction.

  1. “ranging from..” – what ranging?

Accordingly with the reviewer’s comment, we rewrote the sentence to make it clearer (see lanes 63-64).

  1. “14”?

14 is the correct reference of Messaoudi et al.

  1. Guo-Wei at al ..[21] - this reference refers to Xiang, P et al. not Guo-Wei et al.

Thank you for the comment. The correct name is Wang and the correct number of reference is 2.

  1. Lines 130: “I. viscosa ethyl acetate and ethanol extracts display different chemical compositions and are quite different from those reported for I. viscosa grown worldwide..” - it is not surprising but where lays this difference? More information should be given;

Thank you for the comment. Reconsidering the phrase we think that it is of no particular significance for the purpose of the review. We therefore thought of eliminating it.

  1. Section between lines 131-141: this section requires clarification, it is confusing; the statements on effectiveness of ethanolic and ethyl acetate extracts on MCF-7 and MDA-MB-231 cell lines are too general;

Accordingly with the reviewer’s comment, we wrote the sentence to make it clearer (see lanes 122-129).

  1. Table 1: there is no need to repeat the name of the plant in the body of the table, this is given in the table caption; and the first column: the references are given so this column should be removed, it does not provide any new information;

Accordingly with the reviewer’s comments, we eliminated the first column of the Table 1 and the name of the plant in the body of the table.

  1. Line 214: “tested alone or in association with cancer cell lines..” - what does it mean?

It is a mistake; the corrected phrase is “tested alone or in association in cancer cell lines..”

  1. Line 219: “a-b-unsaturated” and “a-methylene-g-lactone,..” – please correct;

We corrected the name of compounds.

  1. Lines 276-279: “silencing ROS production” and “apoptosis is associated with intracellular ROS production” - this is contradictory: silencing ROS production and apoptosis caused by ROS production;

Accordingly, with the reviewer’s comment, we rewrote the sentence to make it clearer (see lanes 266-270).

  1. Table 2: the first column: the references are given so this column should be removed, it does not provide any new information;

Accordingly, with the reviewer’s comments, we eliminated the first column of the Table 2

  1. Figure 1: there is no need to write names of compounds using Capital letters;

We modified the Figure 1 as suggested.

  1. Lines 352-353: “induced by cell treatment.” – please remove;

We modified the phrase as suggested.

Round 2

Reviewer 1 Report

The authors adequately addressed the issues, I have no further comments.